# Echocardiographic Patterns of Left Ventricular Diastolic Function in Cardiac Amyloidosis: An Updated Evaluation

**DOI:** 10.3390/jcm10214888

**Published:** 2021-10-23

**Authors:** Silvia Oghina, Wulfran Bougouin, Mounira Kharoubi, Louis Bonnefous, Arnault Galat, Soulef Guendouz, Mélanie Bezard, Fabien Le Bras, Jean-François Deux, Emmanuel Itti, Anissa Moktefi, Pascale Fanen, Emmanuel Teiger, Dania Mohty, Thibaud Damy, Diane Bodez

**Affiliations:** 1French Referral Centre for Cardiac Amyloidosis, Henri Mondor Teaching Hospital, APHP, 1, Rue Gustave Eiffel, 94010 Creteil, France; mounira.kharoubi@gmail.com (M.K.); bonnefous.louis.x@gmail.com (L.B.); arnault.galat@aphp.fr (A.G.); soulef.guendouz@aphp.fr (S.G.); bezard.melanie@yahoo.fr (M.B.); emmanuel.teiger@aphp.fr (E.T.); thibaud.damy@gmail.com (T.D.); diane.bodez@gmail.com (D.B.); 2GRC Amyloid Research Institute, Henri Mondor Teaching Hospital, APHP, 1, Rue Gustave Eiffel, 94010 Creteil, France; fabien.le-bras@aphp.fr (F.L.B.); deuxjf@gmail.com (J.-F.D.); emmanuel.itti@aphp.fr (E.I.); anissa.moktefi@aphp.fr (A.M.); pascale.fanen@aphp.fr (P.F.); 3Amyloidosis Mondor Network, Henri Mondor Teaching Hospital, APHP, 1, Rue Gustave Eiffel, 94010 Creteil, France; 4Cardiology Department, APHP, Henri Mondor Teaching Hospital, 94010 Creteil, France; 5Paris Cardiovascular Research Centre (PARCC), INSERM Unit 970, 56, Rue Leblanc, 75015 Paris, France; wulfran.bougouin@gmail.com; 6Ramsay Générale de Santé, Hôpital Privé Jacques Cartier, 6, Av du Noyer Lambert, 91300 Massy, France; 7IMRB-INSERM U955, Henri Mondor Teaching Hospital, APHP, 1, Rue Gustave Eiffel, 94010 Creteil, France; 8Lymphoid Malignancies Unit, Henri Mondor Teaching Hospital, APHP, 1, Rue Gustave Eiffel, 94010 Creteil, France; 9Radiology Department, Henri Mondor Teaching Hospital, APHP, 1, Rue Gustave Eiffel, 94010 Creteil, France; 10Nuclear Medicine, Henri Mondor Teaching Hospital, APHP, 1, Rue Gustave Eiffel, 94010 Creteil, France; 11Pathology Department, Henri Mondor Teaching Hospital, APHP, 1, Rue Gustave Eiffel, 94010 Creteil, France; 12Genetic Department, Henri Mondor Teaching Hospital, APHP, 1, Rue Gustave Eiffel, 94010 Creteil, France; 13Referral Centre for AL and Cardiology Department, CHU Dupuytren, 16, rue Bernard Descottes, 87042 Limoges, France; dania.mohty@gmail.com; 14King Faisal Specialist Hospital & Research Centre, Heart Centre, Al Faysal University, Riyadh 112111, Saudi Arabia; 15Cardiology Outpatients Unit, Centre Cardiologique du Nord, 32-36 Rue des Moulins Gémeaux, 93200 Saint Denis, France

**Keywords:** cardiac amyloidosis, heart failure, diastolic function, echocardiography, guidelines

## Abstract

Aims: Multimodal imaging has allowed cardiac amyloidosis (CA) to be increasingly recognised as a treatable cause of heart failure with preserved ejection fraction, but its prognosis remains poor due to late diagnosis. To assess the left ventricular diastolic function (LVDF) patterns in a large contemporary CA cohort according to the current recommendations and to identify their determinants. Methods and Results: We conducted a monocentric, observational study on a cohort of CA patients from a tertiary CA referral centre. Diastolic function was analysed using standard echocardiography and clinical, laboratory and survival parameters were collected. Four hundred and sixty-four patients with one of the three main type of CA were included: 41% had grade III diastolic dysfunction (restrictive mitral pattern), 25% had grade II diastolic dysfunction, and 25% had grade I diastolic dysfunction; 9% were unclassified. No difference was found between the main CA types. After multivariate analyses, grades II and III were independently associated with dyspnoea, elevated NT-proBNP, cardiac infiltration and systolic dysfunction (global longitudinal strain). Grade I patients had a better prognosis. Conclusions: All LVDF patterns can be observed in CA. One quarter of CA patients have grade I LVDF, reflecting the emergence of earlier stage-related phenotypes with a better prognosis.

## 1. Introduction

Cardiac amyloidosis (CA) is an increasingly recognised cause of heart failure rather than a specific aetiology of rare restrictive cardiomyopathies [1,2,3]. Increased left ventricular (LV) filling pressure and restrictive mitral pattern with E/A ratio >2 characterises the restrictive phenotype [4,5]. Echocardiography is an essential part of CA screening, remaining the first-line, routine tool that generates the diagnostic hypothesis and prompts further investigations. Recent expansion of non-invasive CA evaluation, with numerous advanced imaging options, makes earlier recognition possible [6]. Accordingly, the CA landscape has dramatically changed during this period especially as treatments are now available for all CA types [7,8]. Furthermore, the European Association of Cardio-Vascular Imaging (EACVI) and the American Society of Echocardiography (ASE) updated guidelines for evaluating LV diastolic function (LVDF) [9]. To our knowledge, these guidelines have not been applied to describe LVDF in CA. We aim to describe LVDF in a large cotemporary cohort of CA according to these recent guidelines.

## 2. Materials and Methods

### 2.1. Patient Cohort 

We conducted a monocentric, observational, retrospective study based on prospectively collected data at the French referral centre for CA (Henri Mondor Hospital, Creteil, France). All patients with confirmed amyloidosis (light chain (AL), hereditary transthyretin (ATTRv) or wild-type transthyretin (ATTRwt)), with cardiac involvement and available echocardiographic data were included. Biomarkers, multimodality imaging, and biopsies when required were used to confirm the diagnosis as previously described [10,11]. Patients gave informed consent for anonymous publication of scientific data. This study complied with the 1975 Declaration of Helsinki and was approved by our local ethics committee (Créteil) and the French Comité National de l’Informatique et des Libertés (CNIL number 1431858). Data collection was approved by DIRC Ile de France (DC 2009-930).

Clinical (weight, height, blood pressure, heart rate), electrocardiographic (rhythm, low voltage pattern) and laboratory parameters (sodium, troponin and NT-proBNP levels) were recorded at diagnosis. Amyloidosis severity was assessed using the Mayo Clinic score [12] for AL and the specific staging score for ATTR [13].

### 2.2. Echocardiography Evaluation

Echocardiograms were performed and analysed by cardiologists blinded to clinical data using the Vivid 7 system (GE Vingmed, Horten, Norway). LV dimensions and interventricular septal thickness (IVST) were measured according to ASE recommendations from the parasternal long-axis view and were used to calculate LV mass index (LVMi) [9]. LV ejection fraction (LVEF) was calculated using the Simpson’s biplane method. Early (Em) and Late (Am) transmitral diastolic peak flow velocities were measured with pulsed-wave Doppler recordings, and diastolic septal and lateral mitral annulus velocities with tissue Doppler imaging from the apical four-chamber view. Their mean (e’) and the ratio between early diastolic mitral peak flow velocity and e’ (E/e’) were calculated. Mean peak systolic velocity was calculated from mean lateral and septal peak systolic velocity (s’). Left atrial volume was calculated using 2D four and two-chamber apical views. Tricuspid regurgitation systolic jet velocity was measured with continuous wave Doppler. LV peak systolic global longitudinal strain (GLS) was computed offline from the standard three LV apical views using 2D speckle tracking analysis through automated function imaging (AFI, EchoPAC version 203, GE healthcare). Patients with missing data on echocardiograms, significant mitral valve disease or prosthetic mitral valve were excluded from analysis. 

### 2.3. Left Ventricular Diastolic Function Grading

Diastolic function grade was assessed following the ASE/EACVI recommendations for the Evaluation of Left Ventricular Diastolic Function by Echocardiography [9], using algorithm B for patients with myocardial diseases from the guidelines. Patients with atrial fibrillation were evaluated with a specific algorithm. In these guidelines, grade I diastolic dysfunction corresponds to normal left atrial pressure and grades II and III correspond to elevated [9]. The E/e’ ratio was also used as a marker of increased LV filling pressure (if >14 in the guidelines) regardless of the transmitral pattern. 

### 2.4. Follow-Up

Follow-up began after echocardiographic evaluation. An “event” was defined as a composite of death, heart transplant or circulatory assist device. Event status and date were obtained from the usual patient follow-up visits, patient phone calls, or medical records. 

### 2.5. Statistical Analyses

Continuous variables were reported as their mean +/− SD or median (interquartile range) depending on distribution, and categorical variables were expressed as numbers (percentages). Continuous variables were compared using Student’s *t*-test or the Kruskal–Wallis test and categorical variables were compared using the χ2 test. We checked the linearity of continuous variables using fractional polynomial regression, and dichotomised non-linear continuous variables using their median (LVMi, NT-proBNP, and Troponin). We performed univariate analysis to identify the LVDF determinants using a binary outcome: grade I diastolic dysfunction (non-elevated filling pressure) was compared to grades II and III (elevated). Variables associated with LV diastolic dysfunction with *p* value < 0.15 in univariate analysis were included in a multivariate logistic regression model. We assessed collinearity between IVST and LVMi, and between LVEF and GLS and only included IVST and GLS in the multivariate analysis model. *p* values less than 0.05 were considered statistically significant. Uni- and multivariate analyses were performed globally, and for each CA type (AL, ATTRv, and ATTRwt). Missing data were handled using case-complete analysis. Cumulative event curves were plotted using the Kaplan–Meier survival method according to diastolic function grade. Differences between curves were tested by log-rank analysis. Survival was censored at last contact with the patient. Ten patients were excluded as they were lost of follow-up after the first diagnostic assessment. Statistical analyses were performed using STATA software version v14.0 (Lakeway Drive, TX, USA).

## 3. Results

Of the 584 AL and ATTR patients referred to our center between September 2008 and June 2017, 464 were included: 157 AL (34%), 176 ATTRwt (38%) and 131 ATTRv (28%) (Figure 1).

Appendix A shows the CA cohort characteristics. Appendix A shows echocardiographic characteristics. Of 371 patients in sinus rhythm, E/A ratio was ≤0.8 (impaired relaxation) for 75 (20%) and ≥2 (restrictive pattern) for 171 (46%). Notably, eight patients in sinus rhythm (*p* wave on the electrocardiogram) did not have any echocardiographic A wave (atrial electrical-mechanical dissociation). 

LVDF grading for these 371 patients in sinus rhythm according to algorithm B is depicted by Figure 2a: briefly, 92 (25%) had grade I LVDF, 68 (18%) grade II, 168 (45%) grade III and 21 (6%) were unclassified. Among grade I LVDF, 30/92 (33%) had wall thickening below cut-off for hypertrophic cardiomyopathy (<15 mm), and preserved LVEF, meaning they may have been considered cardiomyopathy free after routine first-line echocardiography. Fourteen (46%) of them had AL. 

### Multiparametric Approach Was Made on 4 Criteria, except for One Patient with Missing E/e’ Ratio

LVDF was graded according to a specific algorithm for 93 patients with atrial fibrillation, revealing 23 (25%) with grade I LVDF (Figure 2b). 

Altogether, LVDF was graded for 421 (91%) patients: 115 (25%) grade I, 118 (25%) grade II, and 188 (41%) grade III (Central Figure). 

Table 1 shows baseline clinical and echocardiographic data according to LVDF grade. There was no difference between the three amyloidosis types whereas LVDF varied in symptoms (NYHA class), morphology (IVST), systolic function (LVEF, GSL, and mitral s’) and haemodynamic finding (cardiac index). Clinical, laboratory and echocardiographic parameters were more severely altered for grade II and III patients than grade I. Mean E/e’ was >14 for 28/115 (24%) grade I LVDF, versus 103/118 (87%) and 165/188 (88%) for grades II and III LVDF, respectively. 

Figure 3 displays the gradually increasing prevalence of key abnormal LV echocardiographic parameters of cardiac thickening (IVST > 15 mm) and LV dysfunction (LVEF < 50%, GLS > −16%, Cardiac Index < 1.8 L/min/m^2^, mitral s’ < 8/cm/s) according to LVDF grade from grade I to III. Of note, s’ velocity was the most frequently altered LV parameter, whatever the LVDF grade, seen in 92% of the patients in Grade I, 98% in Grade II and 99% in Grade III, respectively.

Multivariate analysis showed that factors independently associated with elevated LV filling pressures (LVDF Grade II–III vs. I) were NYHA class, LV septal thickness, GLS and NT-proBNP level (Appendix A). Regarding the different amyloidosis types, grade II–III LVDF determinants were respectively: NT-proBNP and NYHA class (III-IV) in AL; NYHA class (III–IV), IVST and troponin level in ATTRv; atrial fibrillation, IVST and GLS in ATTRwt (Appendix A). 

Median follow-up of patients without an event was 26 (16–39) months and 168 (36%) patients experienced an event: 154 (33%) patients died (57 AL, 43 ATTRv, 54 ATTRwt), 11 received a heart transplant (7 AL, 4 ATTRv) and 3 had circulatory support (3 AL). Figure 4 depicts the Kaplan–Meier survival curves according to LVDF grades, showing a significantly different prognosis from grade I to III. 

## 4. Discussion

This is the first study assessing the LVDF grades, defined by ASE/EACVI updated guidelines, in a large and contemporary cohort of CA patients. We showed that less than half of the study population (41%) had a restrictive mitral pattern (grade III LVDF) and a quarter (25%) had grade I LVDF, with significantly different prognoses. Four determinants of grades II and III LVDF were identified: systolic dysfunction, degree of myocardial infiltration, NT-proBNP plasma level and NYHA class. 

### 4.1. LVDF Grade Distribution in CA 

Less than half of the study population had a restrictive mitral pattern, yet CA is still perceived as the archetypal restrictive cardiomyopathy, characterised by an E/A ratio >2 [14]. Indeed, the latest expert consensus on CA diagnosis presents this restrictive pattern or a grade ≥II LVDF as important criteria to suggest the presence of CA, despite recent extensive and multimodal CA characterisation [6,15]. However, cardiologists should consider CA prior to this stage to avoid delayed diagnosis, still common despite better disease knowledge by the cardiologists [16]. Applying the ASE-EACVI recommendations, we found that up to 25% of the study population had grade I LVDF with likely normal LV filling pressures [17]. Indeed, significantly fewer grade I patients had a high E/e’ ratio compared to the other LVDF grades. Furthermore, E/e’ ratio has recently been proposed as one of the criteria used to build echocardiographic diagnostic scores for CA [18]. This being supported by data since 1989, that have shown a continuum of LVDF abnormalities in CA [5]. A previous longitudinal study of 233 patients showed that a restrictive pattern was more frequently encountered in AL patients (55%) than in TTR-CA (36%) [19]. Interestingly, our study showed a similar distribution of LVDF grades in the three CA types and that the amyloidosis type was not an independent LVDF determinant. This suggests that LVDF grade classification cannot solely indicate CA type, and that the recent major improvements in CA imaging may have modified CA presentation at diagnosis, with various types of diastolic filling abnormalities and more often mild diastolic dysfunction.

### 4.2. Structural LVDF Determinants

In our study, systolic dysfunction assessed by LVEF or GLS and the degree of infiltration assessed by IVST or LVMi were independent LVDF determinants. No relationship was observed between age and LVDF grades despite a wide age range in the studied population, which is in contrast with a previous study in the general population [20]. This indicates that the usual diastolic function determinants are not relevant to CA, in accordance with another study on 172 CA patients [21]. Historically, CA was described as “stiff heart” syndrome, and LV stiffness resulting from LV infiltration is linked to the diastolic dysfunction. GLS is another parameter correlated to amyloid infiltration [11]. It is an accurate tool to detect systolic dysfunction, as it assesses LV longitudinal function earlier than LVEF [22], and is also an increasingly recognised parameter of diastolic function assessment [23]. GLS was kept in the multivariate model rather than LVEF because GLS is an accurate and reproducible parameter to differentiate between CA and other causes of LV hypertrophy [24,25]. 

### 4.3. Amyloid Consequences on Heart Failure Determines LVDF Grades in AL 

However, in the subtype analysis, IVST and GLS were not found to be LVDF determinants in AL, conversely to heart failure markers NYHA and NT-proBNP. This may reflect the consequence of diastolic and systolic function impairment and suggests that light chain deposits have a greater myocardial toxicity, meaning LVDF impairment can occur with fewer deposits [10,11,26]. NT-proBNP is also an independent LVDF determinant, especially in AL. NT-proBNP is a strong prognostic factor in AL amyloidosis [27], reflecting cardiac involvement more accurately than infiltration parameters. In our cohort and others, AL patients are the youngest and generally have a poorer prognosis [28], highlighting the need for better recognition even in the case of less marked phenotypes with a lower degree of cardiac infiltration. Recent expert consensuses have reported which “red flags” should be recognised besides an echocardiographic phenotype to raise suspicion of CA, particularly when there are discrepancies between symptoms and cardiac biomarker levels, extra-cardiac signs, or some specific multimodal imaging findings [6,15,29]. 

### 4.4. LVDF Grade and Atrial Fibrillation in ATTRwt

Atrial fibrillation is a diastolic dysfunction determinant in ATTRwt. This could be due to greater left atrial dilatation in this subgroup subsequent to myocardial thickening, which results in impaired LV compliance [14]. Alternatively, increased myocardial infiltration might be related to the increased atrial infiltration, leading to loss of both atrial compliance and systolic function, and elevating the arrythmia risk [30]. However, as atrial fibrillation is not a diastolic dysfunction determinant in ATTRv, atrial infiltration could also be related to age and/or different depositions from wild-type transthyretin fibrils.

### 4.5. Prognostic Value of LVDF Grading in CA

CA prognosis is worse than other hypertrophic cardiomyopathies [31,32]. Our study is the first to show that impaired diastolic function, assessed according to new guidelines, is associated with a poorer prognosis. This is in accordance with Klein et al. who used a different LVDF classification based on transmitral pulse Doppler only [33], and has been similarly documented in patients with no known cardiomyopathy [34]. The E/e’ ratio, which was significantly different between grade I and II-III LVDF, was also recently shown to be an independent mortality predictor in TTR-CA along with GLS [35]. This further emphasises the need to diagnose patients at grade I LVDF, particularly as effective treatments for all CA types are now available. 

### 4.6. Study Limitations

CA is rare, so our study was retrospective with a monocentric cohort. Currently, this is the largest CA cohort to represent the three main CA types and to evaluate LVDF using the recently updated definition combining multiple parameters. Some standard diastolic parameters were missing (isovolumetric relaxation time, A waves duration, mitral inflow during Valsalva manoeuvre), which could have helped to grade patients with missing criteria. It also could have been of interest to analyse left atrial strain since recent literature showed its good prognostic value and diagnostic accuracy in CA. This echographical parameter might be useful in CA and will be the subject of further analyses [36,37,38]. Moreover, our cohort includes various phenotypical presentations, some with few cardiac symptoms or non-typical echocardiographic findings, which could be attributed to the recent availability of multimodality assessment.

## 5. Conclusions

Less than half of the CA study population had a restrictive mitral pattern whatever the amyloidosis type patients with CA. Furthermore, one quarter presented with grade I LVDF, reflecting the emergence of earlier stage-related phenotypes with a better prognosis, due to well-established multimodality diagnosis criteria.

## Figures and Tables

**Figure 1 jcm-10-04888-f001:**
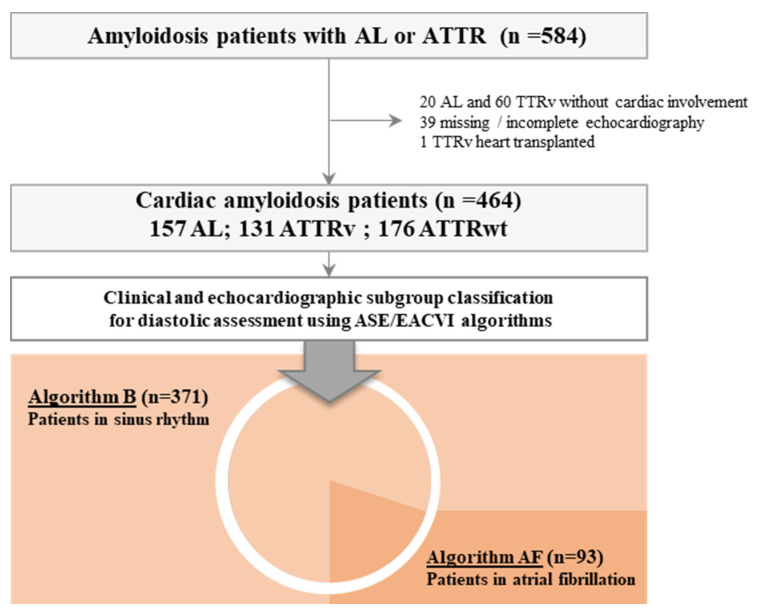
Flow chart. AF: atrial fibrillation; AL: light chain amyloidosis; ATTR: transthyretin amyloidosis, v: hereditary (variant); wt: wild-type.

**Figure 2 jcm-10-04888-f002:**
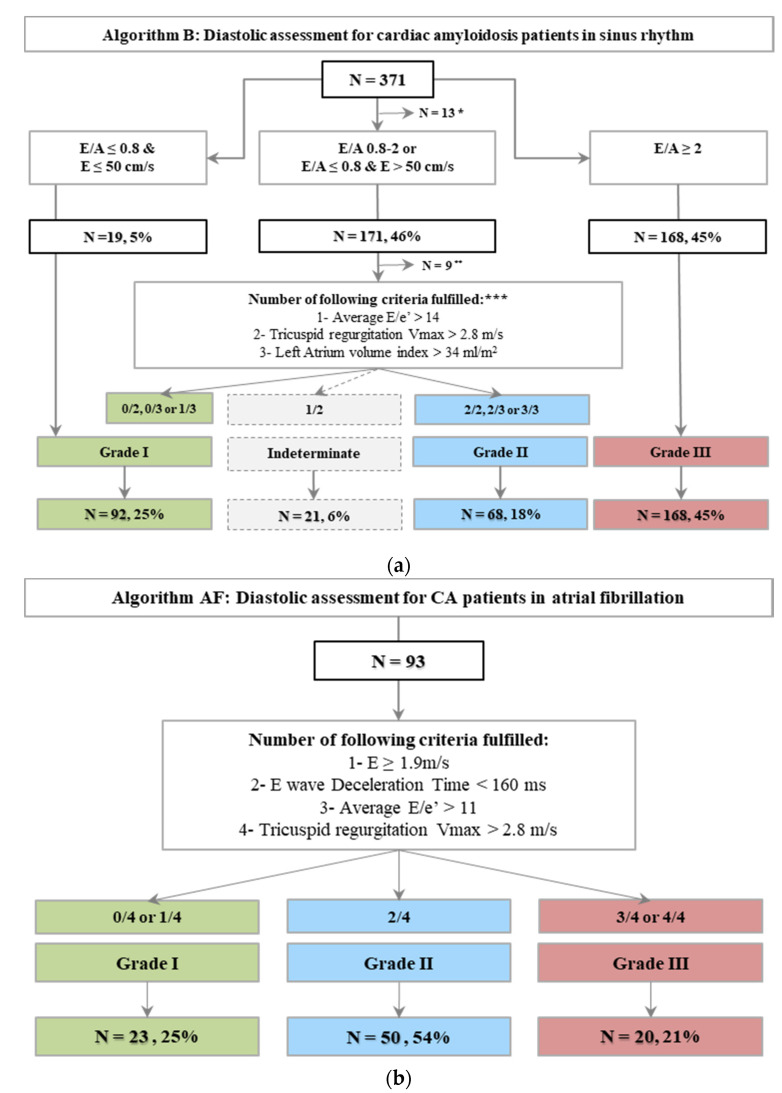
Diastolic function patterns according to LVEF, presence of myocardial disease and rhythm. (**a**) ASE/EACVI Algorithm B: Diastolic assessment for cardiac amyloidosis patients in sinus rhythm. (**b**) ASE/EACVI Algorithm AF: Diastolic assessment for CA patients in atrial fibrillation.* 13 patients without available E/A ratio (5 with no mitral inflow, 8 without A wave despite sinus rhythm), ** 9 patients with insufficient criteria for accurate evaluation of diastolic function, *** if available.

**Figure 3 jcm-10-04888-f003:**
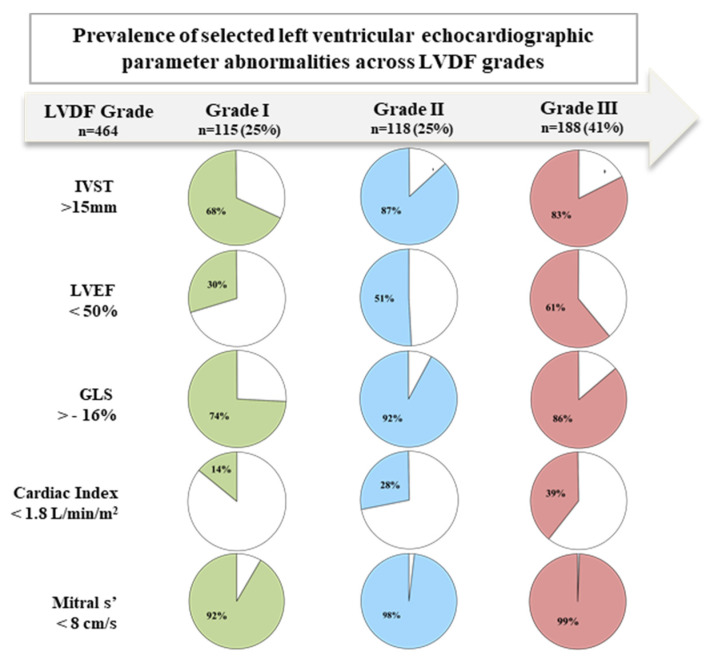
Prevalence of selected echocardiographic parameters abnormalities across left ventricular diastolic function grades. Coloured portions: positive criterion (beyond the cut-off value) GLS: global longitudinal strain; IVST: inter-ventricular septal thickness; LVDF: left ventricular diastolic function; LVEF: left ventricular ejection fraction; s’: systolic pulsed-wave tissue Doppler imaging at the mitral annulus.

**Figure 4 jcm-10-04888-f004:**
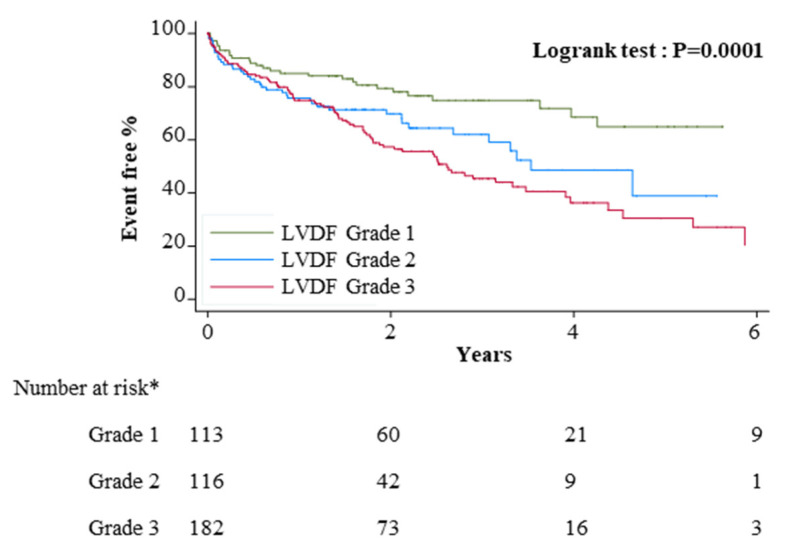
Survival according to the diastolic function grade. * 10 patients excluded patients because of missing follow-up data. Event: death, heart transplant or circulatory assist device LVDF: left ventricular diastolic function.

**Table 1 jcm-10-04888-t001:** Baseline clinical and echocardiographic data according to Left Ventricular Diastolic Function’s grade.

LVDF Grade *	Grade I *n* = 115 (25%)	Grade II *n* = 118 (25%)	Grade III *n* = 188 (41%)	*p*
Amyloidosis type				0.33
AL	43 (31)	37 (27)	59 (42)	
ATTRv	34 (29)	27 (23)	55 (48)	
ATTRwt	38 (23)	54 (32)	74 (45)	
Age, years	72 (64; 80)	77 (69; 83)	75 (64; 80)	**0.006**
Male	82 (71)	84 (71)	131 (70)	0.94
NYHA I–II (vs. III–IV)	35 (32)	63 (55)	103 (56)	**<0.001**
IVST, mm	16 (14; 18)	18 (16; 20)	18 (15; 21)	**0.0001**
LVEF, %	58 (47; 63)	49 (40; 59)	45 (37; 56)	**0.0001**
GLS, -%	12 (8; 16)	10 (7; 13)	9 (7; 13)	**0.0003**
Mean mitral S’ velocity, cm/s	6 (5; 8)	5 (4; 6)	4 (3.5; 5)	**0.0001**
Cardiac index, L/min/m^2^	2.6 (2.1; 3.2)	2.2 (1.7; 2.7)	2.0 (1.6; 2.6)	**0.0001**
LAVI, mL/m^2^	37 (27–54)	48 (41–58)	48 (39–62)	**0.0001**
NT-proBNP, ng/L	2016 (506; 3974)	4510 (2417; 8110)	4372 (2265; 9755)	**0.0001**

Values are median (IQR), or *n* (%). Χ² or Kruskal–Wallis’test. AL: light-chain amyloidosis; ATTRv: hereditary transthyretin amyloidosis; ATTRwt: wild-type transthyretin amyloidosis; LAVI: Left atrial volume index; NYHA: New York Heart Association; IVST: interventricular septal thickness; GLS: global longitudinal strain; LVEF: Left ventricular ejection fraction. * LVDF grade was rated for 421 (91%) patients, in addition to 21 (4%) for whom LVDF grade remains indeterminate, and 22 (5%) for whom echographic parameters were insufficient for evaluation.

## Data Availability

Not applicable.

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
