# Peer review of "Echocardiographic Patterns of Left Ventricular Diastolic Function in Cardiac Amyloidosis: An Updated Evaluation"

_jcm, 2021, doi:10.3390/jcm10214888_

Round 1
Reviewer 1 Report
The manuscript deals with very up to date problem-cardiac amyloidosis and its early recognition,as effective treatments for amyloidosis are now available.The authors aim to describe LVDF in a large contemporary cohort of CA according to the guidelines.
The study is monocentric, observational and retrospective with the rather big study group. The echocardiographic evaluation was performed according to the existing guidelines.The statistical analysis was contemporary and described thoroughly.
The results are presented systematically with tables and figures. However, in the legend of Fig.1 LVEF and IVST should not be included as these words are not mentioned in the figure (or I simply do not see them). It is applausible, that authors pay attention and analise grade I LVDF pts, that 33% of them had wall thickening below cut-off for hypertrophic cardiomyopathy (< 15mm), and preserved LVEF, meaning they may have been considered cardiomyopathy free after routine first-line echocardiography. Another creditable part of the research is that the authors looked for the factors independently associated with elevated LV filling pressures in different amyloidosis types and make suggestions , concerning different determinants of LVDF.
The discussion is very well written. The conclusion is concrete and gives subtle suggestions for future research.
The English level of the manuscript is high. The manuscript will be interesting to the wide auditorium, especially to cardiologists.
A few questions rise, while reading the manuscript: 1).have you noticed any correlations between apical sparing and diastolic function grades? 2).what do you think about atrial strain measurements in pts with amyloidosis (maybe it could help in evaluation of increased infiltration in atrial walls)?
I highly recommend the manuscript for publication, owing to the importance and clinical relevance of the study.
Author Response
We thank the reviewers for their relevant comments, which has helped us further improve our work. A point-by-point reply is provided below.
- Reviewer #1 comment: The results are presented systematically with tables and figures. However, in the legend of Fig.1 LVEF and IVST should not be included as these words are not mentioned in the figure (or I simply do not see them).
We thank the reviewer for this comment. We corrected our legend by removing LVEF and IVST.
- Reviewer #1comment: A few questions rise, while reading the manuscript: 1) have you noticed any correlations between apical sparing and diastolic function grades?
The reviewer raised here a very relevant matter. Relative apical sparing is the subject of many publications. Despite this, our team has already shown that almost 50% of our cohort did not exhibit apical sparing (Reference [11]: Ternacle et al., PMID: 26777222). Moreover, the prevalence of this criterion is highly heterogeneous in the literature (from 62% for Pagourelias et al. in 2017, PMID: 28298286 to 93% for Phelan et al. in 2012, PMID: 22865865). For these reasons, the apex-to-base ratio is not a parameter usually collected in our database. Therefore we cannot provide any correlation analysis with it in this publication. It will be the subject of later publication.
- Reviewer #1 comment: A few questions rise, while reading the manuscript: 2) what do you think about atrial strain measurements in pts with amyloidosis (maybe it could help in evaluation of increased infiltration in atrial walls)?
We thank the reviewer for this comment. Recent literature showed that left atrial strain has good prognosis value (Huntjens et al. in 2021, PMID: 33744146) and good diagnosis accuracy (Brand et al. in 2021, PMID: 32243500, Rausch et al. in 2021, PMID :32728989) in cardiac amyloidosis. Moreover, it is also known that left atrial strain is correlated with left ventricular diastolic function grades in cardiomyopathy free patients and thus might be a useful tool to assess diastolic function (Singh et al. in 2017, PMID: 28017389).
It would have been of great interest to evaluate the correlation between left atrial strain and the degree of diastolic dysfunction as well as its prognostic value on a large cardiac amyloidosis cohort. However, our cohort is retrospective, and we did not have the LA strain analysis package at the time we performed the analysis. These analyses will be the subject of further studies.
- Reviewer #1 comment: I highly recommend the manuscript for publication, owing to the importance and clinical relevance of the study.
We thank the reviewer for his support, as we believe that this publication will raise high interest among the readers of the Journal of Clinical Medicine.
We hope that the new version of our manuscript and our responses have made our article suitable for publication in Journal of Clinical Medicine, and we look forward to receiving your response.
Sincerely,
S. Oghina, MD
Corresponding author

Reviewer 2 Report
The manuscript is well written and focuses on a relevant clinical problem. It is about the presence and severety of diastolic LV dysfunction in patients with cardiac amyloidosis.
I believe, that it is mandatory to add left atrial volume index (LAVI) in the baseline characteristics. This parameter is very relevant in these patients.
Especially, because of a recent publication by Huntjens (doi: 10.1016/j.jcmg.2021.01.016. ) on LA strain in amyloidosis. This should be discussed.
Did the authors perform a multivariate analysis on the role of NT-proBNP in comparision to the diastolic echo parameters? Possibily NTproBNP is the better predictor than the echo-parameters? This should be discussed.
Author Response
We thank the reviewers for their relevant comments, which has helped us further improve our work. A point-by-point reply is provided below.
- Reviewer #2 comment: I believe, that it is mandatory to add left atrial volume index (LAVI) in the baseline characteristics. This parameter is very relevant in these patients.
We thank the reviewer for this comment. We added Left Atrial Volume Index in Table 1 and Table 1 Supp.
Reviewer #2 comment: Especially, because of a recent publication by Huntjens (doi: 10.1016/j.jcmg.2021.01.016.) on LA strain in amyloidosis. This should be discussed.
We thank the reviewer for this comment. Our work focused on a descriptive approach of diastolic function in a cardiac amyloidosis cohort. We included an assessment of the prognosis for each grade of diastolic dysfunction. The reviewer is right in his comment, we could have included the strain of the left atrium for the analysis of the determinants of diastolic function since recent literature shoed that left atrial strain has good prognosis value (Huntjens et al. in 2021, PMID: 33744146) and good diagnosis accuracy (Brand et al. in 2021, PMID: 32243500, Rausch et al. in 2021, PMID: 32728989) in cardiac amyloidosis. Unfortunately, our cohort is retrospective, and we did not have the LA strain analysis package at the time we performed the analysis. These analyses will be the subject of further studies.
We have added this element within the limits of our study, page 10 : “It also could have been of interest to analyse left atrial strain since recent literature showed its good prognostic value and diagnostic accuracy in CA. This echographical parameter might be useful in CA and will be the subject of further analyses.[36-38]”.
- Reviewer #2 comment: Did the authors perform a multivariate analysis on the role of NT-proBNP in comparision to the diastolic echo parameters? Possibily NTproBNP is the better predictor than the echo-parameters? This should be discussed.
We thank the reviewer for this comment. We indeed have performed a multivariate analysis on the role of NT-proBNP in left ventricular filling pressure assessment. Inclusion of echo-parameters in the multivariate analysis would lead to collinearity (considering that outcome, diastolic dysfunction, is defined by echo parameters).
NT-proBNP > 3600 ng/l was an independent factor associated with elevated left ventricular filling pressure. Data are shown in Table 3 Supp. We added “NT-proBNP is also an independent LVDF determinant, especially in AL”, before “NT-proBNP is a strong prognostic factor in AL amyloidosis,[27] reflecting cardiac involvement more accurately than infiltration parameters.” in the discussion, page 9.
- Reviewer #2 comment: The manuscript is well written and focuses on a relevant clinical problem. It is about the presence and severety of diastolic LV dysfunction in patients with cardiac amyloidosis.
We thank the reviewer for this comment.
We hope that the new version of our manuscript and our responses have made our article suitable for publication in Journal of Clinical Medicine, and we look forward to receiving your response.
Sincerely,
S. Oghina, MD
Corresponding author
